# A multi-state dynamic process confers mechano-adaptation to a biological nanomachine

Navish Wadhwa [1,2,5] ✉, Alberto Sassi [3,5], Howard C. Berg[4] & Yuhai Tu [3] ✉

Adaptation is a defining feature of living systems. The bacterial flagellar motor adapts to changes in the external mechanical load by adding or removing torque-generating (stator) units. But the molecular mechanism behind this mechano-adaptation remains unclear. Here, we combine single motor eletrorotation experiments and theoretical modeling to show that mechano-adaptation of the flagellar motor is enabled by multiple mechanosensitive internal states. Dwell time statistics from experiments suggest the existence of at least two bound states with a high and a low unbinding rate, respectively. A first-passage-time analysis of a four-state model quantitatively explains the experimental data and determines the transition rates among all four states. The torque generated by bound stator units controls their effective unbinding rate by modulating the transition between the bound states, possibly via a catch bond mechanism. Similar force-mediated feedback enabled by multiple internal states may apply to adaptation in other macromolecular complexes.

Through billions of years of evolution, living organisms have devised myriad ways to move around[1]. For single-celled bacteria, one of the most common mode is swimming in aqueous environments by the rotation of thin helical filaments called flagella[2–4]. Flagellar rotation is powered by an intricate nanomachine, the flagellar motor (diameter ~ 50 nm), a macromolecular complex assembled from more than 20 types of proteins through a highly regulated process[5,6]. The core structure of the motor consists of a rotor that traverses the cell envelope, and an inner-membrane embedded stator that surrounds it and generates force[7–9]. The stator is made up of individual units that anchor in the peptidoglycan layer of the cell wall and dissipate ion motive force to generate rotation[10–15].

A central feature of living systems is their ability to adapt to changes in their environment. Previous work on adaptation has largely focused on biochemical networks in sensory systems, such as the bacterial chemotaxis signaling pathway[16]. Much less is known about how macromolecular complexes adapt to changes in their mechanical environment. The bacterial flagellar motor has emerged as a canonical example of an adaptive macromolecular complex[17,18]. An

important recent discovery is that instead of being static once assembled, the motor is a dynamic complex that continuously adapts to external mechanical demand by remodeling itself[19,20]. Under high load, the motor adds force-generating stator units and thus increases its output (and vice versa)[21–24], by tuning the dynamic turnover of the stator units between a motor-bound set and an inner membrane-embedded pool[25,26]. The goal of this study is to elucidate the physical mechanism underlying mechano-adaptation of the motor.

Previous work modeled the assembly of a stator unit into the motor as a two-state process, with a bound ("on") and an unbound ("off") state. The rate of transition from the bound to unbound state decreases with increasing motor torque in response to higher load[21–23,27]. While this simple model captures population-averaged kinetics, its broader validity has not been thoroughly tested. Indeed, recent work by Perez-Carrasco and co-workers found that the asymmetry observed in the timescales of relaxation to steady state (in population-averaged data) from either a large or a small number of stator units could not be explained by a simple, two-state model[28]. This gap could be best resolved by the introduction of a third, strongly

[1]Department of Physics, Arizona State University, Tempe, AZ 85287, USA. [2]Biodesign Center for Mechanisms of Evolution, Arizona State University, Tempe, AZ 85287, USA. [3]IBM T. J. Watson Research Center, Yorktown Heights, NY 10598, USA. [4]Department of Molecular and Cellular Biology, Harvard University, Cambridge, MA 02138, USA. [5]These authors contributed equally: Navish Wadhwa, Alberto Sassi. ✉e-mail: Navish.Wadhwa@asu.edu; yuhai@us.ibm.com

bound state into the model[28]. Similarly, by analyzing the steady state distribution of the time intervals between binding and unbinding events in single motors, Shi et al. discovered the existence of an additional short-lived "hidden" state in which the stator unit transiently disengages from the rotor while remaining assembled in the motor[29]. These examples suggest that, to gain a deeper understanding of the mechanisms that enable adaptive stator remodeling, we must go beyond population-averaged data and analyze the statistics of single binding and unbinding events in individual flagellar motors. The motor behavior must be described not just in steady state but also during the adaptation process.

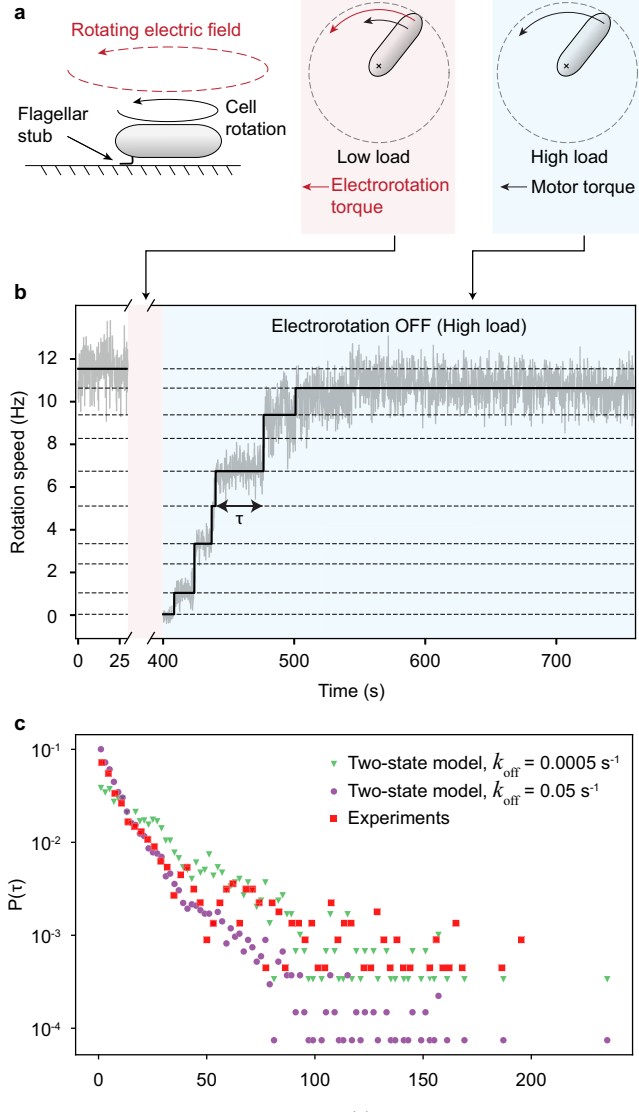

**Fig. 1 | Single-motor electrorotation experiments. a** Experimental strategy. The cell is tethered to a surface via a short flagellar stub (left). A high-frequency rotating electric field then applies an external torque (red) on the cell, which decreases the motor load (middle). Under low load, the motor releases its bound stator units. The electrorotation torque is then switched OFF, which increases the load on the motor (right). In response, the motor recruits stator units, leading to stepwise increases in rotation speed. **b** Motor speed (gray) as a function of time in an electrorotation experiment, showing fitted steps (black). The dashed lines indicate the discrete speed levels. **c** Distribution of the dwell times $\tau$ from the electrorotation experiments (red squares), as well as from simulations of two different scenarios of the two-state model ($k_{off}$ = 0.0005 s$^{-1}$, green triangles; $k_{off}$ = 0.05 s$^{-1}$, purple disks). Bin size $d\tau$ = 3 s.

Here, we conduct a detailed analysis of remodeling events in single flagellar motors of *Escherichia coli* as they adapt to a sudden increase in load. From the distribution of dwell times of these events, we find clear signatures of additional states beyond the simple bound and unbound states. Motivated by these observations, we propose a new model with four states (unbound, loosely bound, tightly bound, and transiently unbound). This treatment provides a coarse-grained approximation for a more complete model in which a bound stator unit can occupy a continuum of states with different dissociation rates. We present a theoretical treatment of the coarse-grained model using first-passage-time analysis, which allows us to calculate key statistics for the dwell time analytically. Our coarse-grained model demonstrates excellent quantitative agreement with the dwell-time statistics extracted from experimental data. Finally, we discover new features of the on-process, whereby the on-rate of an incoming stator unit is a non-monotonic function of the number of previously bound units. Put together, this work reveals that the mechanosensitive remodeling of the flagellar motor is powered by molecular interactions that are more nuanced and complex than previously thought.

## Results

### Single-motor electrorotation experiments

To accurately measure the dynamics of flagellar motors during the adaptation process, we conducted a large number of single-cell experiments with the bacterium *Escherichia coli*. We tethered individual *E. coli* cells to a surface via a short flagellar stub, causing the motor to rotate the cell body instead of the flagellar filament and thus operate under a very high load (Fig. 1a). In the initial phase of the experiment, we observed the cell rotation for 30 s. We then applied a high frequency rotating electric field to the cell, which exerted a large external torque on it in the same direction as the motor torque. This greatly reduced the load acting on the motor and stimulated mechano-adaptation in the motor via the release of bound stator units. We kept electrorotation ON for a total of 6 min. At the end of 6 min, we turned electrorotation OFF, which suddenly increased motor load. Once again, the motor adapted, this time by the sequential addition of stator units, with the motor speed increasing in a stepwise fashion over time (Fig. 1b). This phase of the experiment also lasted for 6 min, during which we made continuous measurements of the motor speed at a high temporal resolution.

The time trace of the motor output (i.e., rotation speed) consists of discrete levels that correspond to different number of stator units driving the motor (Fig. 1b). We used a step-fitting algorithm (see "Methods") to extract these discrete levels from the noisy speed data at each time instant (black line in Fig. 1b). Once the levels were identified, we associated them with a corresponding number of stator units, assuming that when the speed is close to 0 Hz no units are bound to the rotor. Each new higher level is then due to the addition of a single unit. To validate our analysis pipeline, we plotted the entire set of extracted speed levels against the number of stator units associated with those levels (Supplementary Fig. 1a). The approximately linear dependence of rotation speed on the number of bound stator units is consistent with the generally accepted view that, under high load, each stator unit supplies the same torque, resulting in uniform spacing between successive speed levels[25,30]. In addition, we plotted the distribution of the maximum number of stator units ($N_{max}$) observed during the remodeling process (Supplementary Fig. 1b). In agreement with previous observations in *E. coli*, we find that the number of stator units bound to the motor is at most 11[31]. The observed variation in $N_{max}$ could be caused by insufficient observation time or different packing arrangements of stator units around the circumference of the rotor.

The simplest possible model for the molecular interactions underlying stator remodeling is one in which a stator unit can exist in

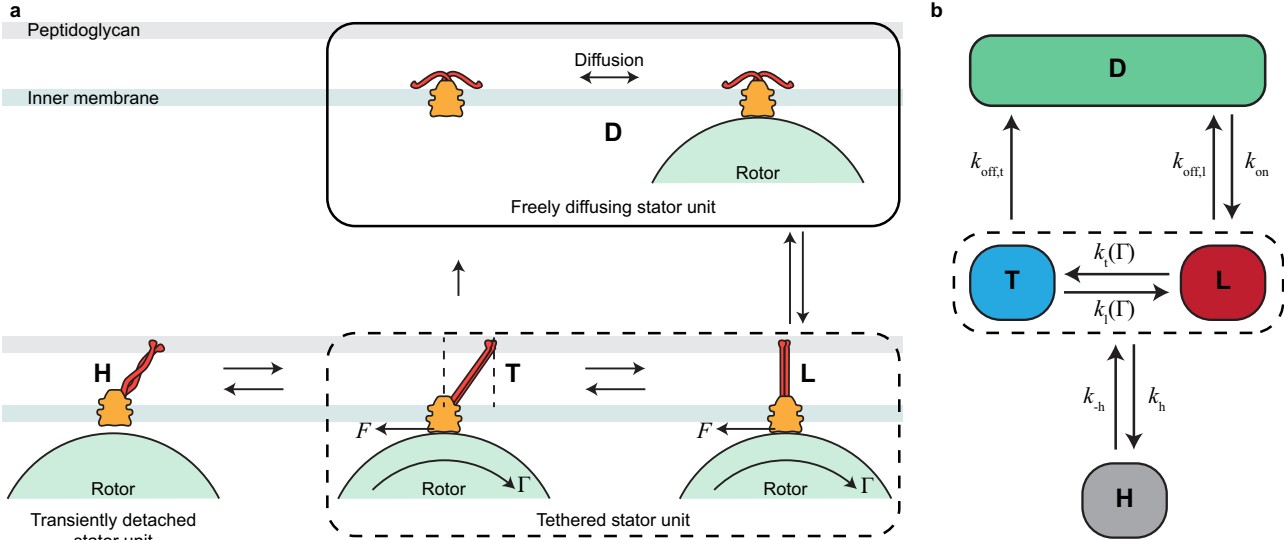

**Fig. 2 | A multi-state model for adaptive stator remodeling. a** An unbound stator unit is in a closed state (D) in which it freely diffuses within the inner membrane and does not translocate protons. Upon interacting with a rotor of radius $R$, the stator unit transitions into a loosely bound state (L). In this state, the protons start flowing through the stator unit, resulting in the production of a torque $\Gamma$ on the rotor. As a reaction, a force $F = \Gamma/R$ acts on the stator unit, which displaces it in the direction of the force. This displacement away from the point of tether lowers the off rate and results in a tightly bound state (T). A bound stator unit can occasionally detach from the rotor, resulting in a transiently unbound "hidden" state (H). **b** A simplified 4-state model of stator dynamics. A stator unit can go between the diffusive state (D) and the loosely bound state (L) with an on rate, $k_{on}$ and an off rate $k_{off,l}$. From the L state, it can go to the tightly bound state (T) with a rate $k_t$ and back with a rate $k_l$, both of which depend on the torque ($\Gamma$). From the T state, it can transition to the D state at a much lower off rate $k_{off,t}$. This model also includes the transiently unbound hidden state (H). The H state couples with either the T or the L state with rates $k_h$ and $k_{-h}$.

one of only two states—an unbound state, in which the stator unit diffuses freely in the inner membrane, and a bound state, in which it interacts with the rotor to generate torque. Indeed, this simple model has successfully described the population-averaged kinetics of stator remodeling and its dependence on experimental parameters, such as load[21–23]. To further test its validity, we simulated two different scenarios (with different dissociation rates $k_{off}$) of the two-state model for the entire stator complex. We compared distributions of dwell times for these simulations with the experimentally observed distribution (Fig. 1c). We find that neither of the two scenarios of the two-state model fully agree with the experimental data. While the two-state model with a larger $k_{off}$ (= 0.05 s$^{-1}$) is consistent with the experiments for small dwell times ($\tau \leq 50$ s), it lacks the long-time tail ($\tau \geq 50$ s) observed in the experiments. Conversely, the two-state model with a smaller $k_{off}$ (= 0.005 s$^{-1}$) does generate a long-time tail consistent with the experiments, but it disagrees with the experimental distribution for short dwell times. In Supplementary Note 2, we describe several other disagreements between the two-state model and our experiments with respect to dwell time statistics. Taken together, these disagreements between the two-state model and our experimental data suggest the existence of additional states that introduce additional timescales to the distribution of dwell times.

**A mechanistic model for stator dynamics**
Motivated by the multiple timescales observed in the distribution of dwell times, we propose a minimal model for stator binding that provides a mechanistic understanding of stator dynamics. Our model is based on the assumption that the affinity between the rotor and a single stator unit is not constant. As a simple approximation of this assumption, we define two separate states for each bound stator unit, characterized by different rates of unbinding. The addition of these internal states leads to the emergence of additional time scales in the distribution of dwell times, in line with the experimental evidence.

Figure 2a illustrates the key elements of our model. Away from the motor, a stator unit is in the diffusive state (D) in which it floats freely in

the inner membrane and the proton channels inside it are closed. When a diffusing stator unit collides with a rotor, it may transition to a bound state (Fig. 2a), in which it is tethered to the peptidoglycan (PG) layer of the cell envelope and the proton channels open. Subsequently, protons flow through the stator unit and generate torque ($\Gamma$) that drives the rotation of the rotor, which has a radius $R$. Newton's third law dictates that a counter force $F = \Gamma/R$ acts on the stator unit in the opposite direction (Fig. 2a). This counter force moves the tethered stator unit away from its landing point by a displacement that depends on the force $F$, and thus on the torque $\Gamma$. We hypothesize that the rate at which the stator unit unbinds from the rotor decreases with increasing displacement from the landing point (see Supplementary Note 1 for details). To simplify the problem, instead of modeling a continuous displacement that depends on the torque, we use a coarse-grained approximation (Fig. 2b) to describe the internal state of a bound stator unit as being in one of two possible states: a loosely bound (L) state with a higher unbinding (off) rate $k_{off,l}$, and a tightly bound (T) state with a lower off rate $k_{off,t} \ll k_{off,l}$. The loose state (L) corresponds to a stator unit with a small displacement; the tight state (T) represents a stator unit with a large displacement. Despite being identical in terms of torque production, the difference in their displacement leads to different off rates. This enables the flagellar motor to undergo load-dependent remodeling, as described next.

This coarse-grained model has two important features that dictate its behavior. The first is that, when a freely diffusing stator unit in the D-state binds the motor, it enters the loose state (L) by definition, at a rate $k_{on}$. The on rate for the entire complex, $k_+$, depends upon the number of available sites ($N_{tot} - N$), where $N_{tot}$ is the total number of sites and $N$ is the number of occupied sites. It can additionally depend on other parameters, such as the motor speed $\omega$, which is linearly proportional to the number of attached stator units $N$ (Supplementary Fig. 1a). The second important feature of this model is that the transition rates ($k_l$ and $k_t$) between the loose state (L) and the tight state (T) depend on the torque ($\Gamma$). In particular, $\frac{k_t}{k_l}$ increases with torque so that the equilibrium probability of a bound stator unit to be in the tight

state $P_t = k_t/(k_t + k_l)$ increases with torque. Consequently, the effective (observed) off rate $\tilde{k}_{off} = P_t k_{off,t} + (1 - P_t)k_{off,l}$, which is averaged over the two bound states ($T$ and $L$), is smaller at higher torque because the tight state has a much smaller off rate $k_{off,t}(\ll k_{off,l})$, which is set to zero ($k_{off,t} = 0$) hereafter for simplicity. Thus, torque controls the effective off rate by controlling the internal dynamics of the bound stator units.

While the above described states (D, L, and T) may be sufficient to describe the on process from freely diffusion unbound stator units and the torque dependence of the off process, our data as well as previous work[29] suggest that there is an additional "hidden" unbound state different from D. Specifically, a bound stator unit can occasionally become detached from the rotor at a rate $k_h$. The detached unit quickly attaches again to the rotor with a much higher rate $k_{-h} \gg k_h$, resulting in this hidden state being very short lived. To account for this finding, we introduce this hidden state (H) into our model (Fig. 2b). The H-state is an unbound state (it does not exert force on the rotor). However, it is not the same as the diffusive state (D) because the stator unit stays in the vicinity of the rotor and quickly rebinds with the rotor and becomes attached again. The short-lived H state is evidenced by the fastest decay time scale in the observed distribution of dwell times (leftmost data points in Fig. 1c).

Altogether, we have the minimal 4-state (D-L-T-H) model that describes the dynamics of stator assembly (Fig. 2b). Aside from the rates ($k_{\pm h}$) related to the fast transient state (H), the model is defined by four rate parameters: $k_{on}$, $k_{off,l}$, $k_t$, and $k_l$. The on rate $k_{on}$ can additionally depend on $N$ because of its dependence on the rotation speed $\omega$, which is linearly proportional to $N$ (Supplementary Fig. 1a). These important biophysical parameters are hard to measure directly. In the following, we determine their values from a statistical analysis of the remodeling data obtained from single flagellar motors during their adaptation to a sudden increase in load.

### The statistics of dwell times and first-passage-time analysis

Due to a separation of timescales, i.e., the transition rate $k_{-h}$ from the H state to the bound state being larger than the other rates, we simplify our analysis by first identifying the H states in the time series using a short time-scale threshold for the duration of the H states (Supplementary Fig. 2). From the H states identified in the experimental data, the kinetic rates ($k_{\pm h}$) to and from the H state can be determined.

Once the short-lived H states are identified, we can separate them from the rest of the time series and focus on analyzing the transitions between the other three states (L, T, and D), which enable mechano-adaptation in the motor. An $N$-stator state ends with either an increase or decrease in $N$ ($N \to N \pm 1$). The stochastic dynamics of $N$ are controlled by both the on ('+') and off ('−') processes, and whether the '+' or the '−' transition is observed depends on which of the two happens first. Thus, the statistics of dwell times can be understood as a first-passage-time (FPT) problem[32] with two independent stochastic processes ('+' and '−'). Mathematically, when the number of stator units reaches $N$ at time $t = 0$, we can compute the survival probability $S(t)$, i.e., the probability the motor stays in state-$N$ at a later time $t \geq 0$. The distribution $P(\tau)$ for dwell time $\tau$ can then be determined from $S(t)$: $P(\tau) = -\frac{dS}{dt}\big|_{t=\tau}$.

In our model, the on process has a time-independent rate $k_+ = (N_{tot} - N)k_{on}(N)$, where $k_{on}(N)$ is the on rate for a single stator unit. However, the off process has a rate $k_-(t|N)$ that is time-dependent because of the existence of the two bound states and their different off rates. This can be understood by considering a unit-$i$ ($i = 1, 2, \ldots, N$) that binds to the rotor at time $t_i(\leq 0)$ in the L state. Its survival probability $S_i(t)$ can be determined analytically:

$$S_i(t) = ce^{-\sigma_+(t-t_i)} + (1-c)e^{-\sigma_-(t-t_i)}, \quad (1)$$

where $\sigma_\pm = \frac{1}{2}[(k_l + k_t + k_{off,l}) \pm \sqrt{(k_l + k_t + k_{off,l})^2 - 4k_l k_{off,l}}]$ are the two eigenvalues of the transition rate matrix for the two bound states (T

and L), and $c = \frac{\sigma_+ - k_l - k_t}{\sigma_+ - \sigma_-}$ is a constant (see Supplementary Note 3 for detailed derivation). The off rate for the unit-$i$ can then be determined from $S_i$ for $t \geq t_i$:

$$k_{-,i}(t) = -S_i^{-1}\frac{dS_i}{dt} = \frac{c\sigma_+ e^{-\Delta\sigma(t-t_i)} + (1-c)\sigma_-}{ce^{-\Delta\sigma(t-t_i)} + (1-c)}, \quad (2)$$

where $\Delta\sigma = \sigma_+ - \sigma_-$. Equation (2) shows that $k_{-,i}$ starts with the "bare" off rate $k_{-,i}(t_i) = c\sigma_+ + (1-c)\sigma_- = k_{off,l}$ when the stator unit first becomes bound at $t = t_i$ and decreases to its equilibrium value $k_{-,i}(\infty) = \sigma_-$ at $t - t_i \gg \tau_e$ with $\tau_e = \Delta\sigma^{-1}$ is the timescale to reach equilibrium between the two bound states (T and L).

The overall off rate (summed over all the bound stators), $k_-(t|N) = \sum_{i=1}^{N} k_{-,i}(t)$, thus also depends on time, which indicates a non-Markovian (memory) effect due to the existence of the hidden bound states. Here, the T-state and the L-state are called "hidden" states because we can not tell them apart directly from the experimental observations, i.e., torque (or speed) of the motor. Given the rates $k_+(N)$ and $k_-(t|N)$, the survival probability satisfies:

$$\frac{dS(t|N)}{dt} = -(k_+(N) + k_-(t|N))S, \quad (3)$$

which can be solved with the initial condition $S(0|N) = 1$ to obtain a closed-form expression for the survival probability: $S(t|N) = \exp[-k_+ t]\prod_{i=1}^{N}\frac{S_i(t)}{S_i(0)}$. Details of the FPT analysis and derivation of the solution for $S(t|N)$ are given in the Supplementary Note 3.

### Quantitative comparison between experiments and theory

First, we estimated the transition rates to and from the H state as $k_h \approx N_H/T_{tot}$ and $k_{-h} \approx 1/\tau_H$, where $N_H$ is the number of occurrences of the short-lived H state, $\tau_H$ is their average dwell time, and $T_{tot} = \sum N_i \tau_i$ is a weighted sum of dwell times where each weight is the number of stator units that are already bound. From our experiments, we have $N_H = 43$, $\langle T_{tot}\rangle = 136620$ s, and $\tau_H = 4.1$ s, leading to $k_h \approx 0.0003$ s$^{-1}$ and $k_{-h} \approx 0.24$ s$^{-1}$, which are in excellent quantitative agreement with the rates determined previously by measurements of steady-state motor dynamics[29].

Next, for each $N$, we separated the statistics for dwell times into two distributions: $P_+(\tau|N)$ and $P_-(\tau|N)$, which are the probabilities for observing the $N \to N+1$ and $N \to N-1$ transitions, respectively, for the first time at $t = \tau$, where $t = 0$ is the time when the $N$-state is first reached. The overall distribution of $\tau$ regardless of the end state is $P(\tau|N) = P_+(\tau|N) + P_-(\tau|N)$. The observed histograms for these dwell times (Supplementary Fig. 4) show the occurrence of long dwell times far outside of the exponential distribution with a shorter mean dwell time, which indicates the existence of multiple timescales even for a given $N$. In our model, from the survival probability $S(t|N)$ obtained by solving Eq. (3) for each $N \in [0, N_{tot}]$, we can determine the two dwell-time distribution functions $P_+(\tau|N) = k_+(N)S(\tau|N)$ and $P_-(\tau|N) = k_-(\tau|N)S(\tau|N)$, which can be compared with the observed distribution of dwell times.

From the distribution of dwell times for each $N$, we compute several key statistical properties that characterize the underlying stochastic process:

$$f_\pm(N) \equiv \int_0^\infty P_\pm(\tau|N)d\tau, \langle\tau\rangle(N) = \int_0^\infty \tau P(\tau|N)d\tau, V(N) \equiv \frac{\sigma_\tau^2}{\langle\tau\rangle^2}, \quad (4)$$

where $f_\pm$ is the fraction of '+' (on) or '−' (off) transitions ($f_+ + f_- = 1$); $\langle\tau\rangle(N)$ is the mean dwell time for the $N$-stator state; $V$ is the variance of dwell time $\sigma_\tau^2 \equiv \int_0^\infty [\tau - \langle\tau\rangle(N)]^2 P(\tau|N)d\tau$ normalized by the square of its mean, which measures the magnitude of the variation in dwell times. We can also define $\langle\tau_\pm\rangle(N) = \int P_\pm(\tau|N)d\tau$ as the average '+' or '−' dwell time, and we have: $\langle\tau\rangle(N) = \langle\tau_+\rangle(N)f_+(N) + \langle\tau_-\rangle(N)f_-(N)$.

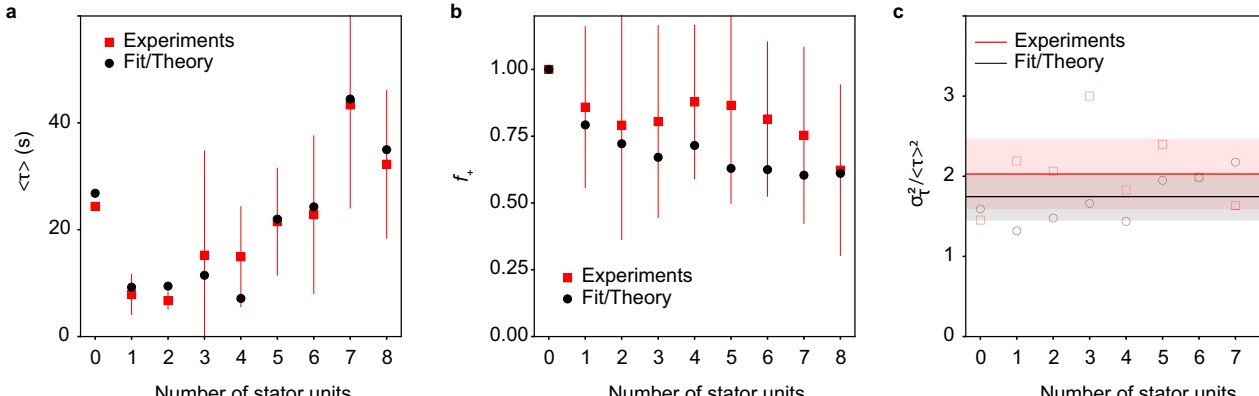

**Fig. 3 | Quantitative comparison between experiments and theory. a** The average dwell time $\langle\tau\rangle$ for each stator number $N$. **b** The fraction of on events $f_+$ for each stator unit number $N$. **c** The normalized variance $V \equiv \sigma_\tau^2/\langle\tau\rangle^2$ versus $N$. The average $V$ over $N$ and the standard deviation of $V$ over $N$ are shown as the solid line and the shaded band for experiments and theory, respectively. In all panels, experimental data are in red and fit/theory results in black. The sample size for 0, 1, 2,... 8 stator units is 74, 63, 75, 70, 77, 81, 80, 83, and 74, respectively.

These characteristic properties correspond to the lower order moments of the distributions of dwell time, which are less susceptible to measurement noise. Therefore, comparisons of these properties between model and experiments lead to robust estimates of the model parameters.

In Fig. 3, we show the comparison between experimental data and our model results for the average dwell time $\langle\tau\rangle$, the fraction of the '+' events $f_+$, and the normalized variance $V$ for different numbers of stator units $N$ (see Supplementary Note 5 for additional details on model fitting). The agreement between the experimental data and our model results supports our theory, which incorporates multiple hidden bound states. This is seen most clearly in the normalized variance $V$ (Fig. 3c). If there were only one bound state, the dwell time distribution would be a single exponential distribution, which would lead to $V = 1$ (Supplementary Note 2). From the experimental measurements, we found $\langle V\rangle = 2.0 \pm 0.4$. The large normalized variance can only be explained by a model with multiple bound states like the one we proposed in Fig. 2, which leads to a theoretical value $\langle V\rangle = 1.75 \pm 0.4$, consistent with experiments (see Supplementary Note 4 for additional discussion on the possible role of cell-to-cell variability).

The quantitative agreement between our model and the experimental data confirms the existence of multiple bound states. Furthermore, it allows us to determine the dominant off rate $k_{\text{off,l}}$ from the L-state and the transition rates $(k_t, k_l)$ between the two bound states ($T$ and $L$), which are not possible to measure directly. In particular, from the fitting of our model to the experimental data (Fig. 3), we determined the values of key model parameters: $c = 0.30 \pm 0.11$, $\sigma_+ = 0.19 \pm 0.1\,\text{s}^{-1}$, and an upper bound for the much lower rate $\sigma_- \leq 0.0005\,\text{s}^{-1}$ (see Supplementary Note 5 for details). In the regime $\sigma_+ \gg \sigma_-$, which is valid at high load, these model parameters are related to $k_t$, $k_l$, and $k_{\text{off,l}}$: $\sigma_+ \approx k_t + k_l + k_{\text{off,l}}$, $\sigma_- = \frac{k_l k_{\text{off,l}}}{\sigma_+}$, and $c \approx \frac{k_{\text{off,l}}}{\sigma_+}$, from which we obtain: $k_{\text{off,l}} \approx c\sigma_+ = 0.057 \pm 0.023\,\text{s}^{-1}$, $k_t \approx (1-c)\sigma_+ = 0.13 \pm 0.07\,\text{s}^{-1}$, and $k_l \approx \sigma_-/c \leq 0.0017\,\text{s}^{-1}$. Thus, the results of our fit show that, at high load, $k_t \gg k_l$, which leads to a much decreased effective off rate $\sigma_- \approx (1 + k_t/k_l + k_{\text{off,l}}/k_l)^{-1} k_{\text{off,l}} \ll k_{\text{off,l}}$. The reason for this significant decrease in the off rate at high load is that, once bound in the L-state, a stator unit quickly transitions to the more stable T-state. The range of equilibrium off rate $\sigma_-$ at high load inferred from our experiments is consistent with those measured in previous experiments[29]. At low load, the equilibrium is shifted towards the L-state, i.e., $k_t \ll k_l$, and our model predicts a much larger effective off rate $\sigma_- \approx k_{\text{off,l}} = 0.057\,\text{s}^{-1}$, which is in excellent agreement with previous measurements at low load[21].

## On rate depends on the number of bound units

Our analysis also provides new information about the on process. Based on the FPT analysis, we have $P_+(\tau|N) = k_+(N)S(t|N)$, which leads to $f_+(N) = \int_0^\infty P_+(\tau)d\tau = k_+(N)\int_0^\infty S(\tau|N)d\tau = k_+(N)\langle\tau\rangle$. Therefore, we can obtain the on rates for different stator number $N$, $k_+(N) = f_+(N)/\langle\tau\rangle(N)$, from the observed $f_+(N)$ and $\langle\tau\rangle(N)$. Furthermore, we can obtain $k_{\text{on}} = k_+/(N_{\text{tot}} - N)$ as the on rate for each empty binding site. In Fig. 4, we plot $k_{\text{on}}(N)$ for different values of $N = 0, 1, \ldots, 8$. This plot reveals several interesting features about the on process. First, the nucleation rate, or the initial on rate for $N = 0$: $k_{\text{on}}(0) \approx 0.0037\,\text{s}^{-1}$ is much smaller than the on rates for $N \geq 1$. Second, the on rates for $N \geq 3$ are roughly the same, with $k_{\text{on}}(N \geq 3) \approx 0.0067\,\text{s}^{-1}$. Third, the on rates for $N = 1, 2$ are higher than the rates for $N \geq 3$. The first two features were also reported in a recent study by Ito et al.[33]. At high load, the motor speed is linearly proportional to $N$ and the low initial on rate has been attributed to a possible dependence of $k_{\text{on}}$ on the rotational speed $\omega$ in a nonlinear, sigmoid-like fashion[33]. However, as far as we know, the enhanced on rates for $N = 1$ and $N = 2$ have not been reported before. These high rates are related to the short average dwell times for $N = 1$ and $N = 2$ (Fig. 3a).

We do not understand the molecular details of how a freely diffusing (unbound) stator is incorporated into a rotating motor. The dependence of $k_{\text{on}}(N)$ on $N$ is likely caused by effect(s) of the motor speed on the on rate. The non-monotonic dependence we found further suggests that there may be multiple counteracting effects. On the one hand, a higher motor speed may enhance both the probability of collision between a diffusive unbound stator unit and the rotor and the collision strength, effects that could contribute to a higher on rate[33]. On the other hand, a higher speed would also mean a shorter contact time, which could lead to a lower on rate, as previously suggested by Wadhwa et al.[22]. By combining these two possible counteracting effects, we use a minimal phenomenological model to fit the data:

$$k_{\text{on}}(N) = k_{\text{on}}(0) \times f_1(N) \times f_2(N), \tag{5}$$

where $f_1(N)(f_2(N))$ is a monotonically increasing (or decreasing) function of $N$ that describes the speed-dependent enhancing (reducing) effect of the on rate with $f_1(0) = f_2(0) = 1$. Here, we chose the simplest forms with the smallest number of parameters for these two functions that are consistent with previous studies[22,33]. As shown in Fig. 4, this simple model yields reasonable agreement with the experimental results, which supports the hypothesis that the motor speed may affect the on process through two opposing mechanisms. While the functional form of $k_{\text{on}}(N)$ is speculative, the agreement between this

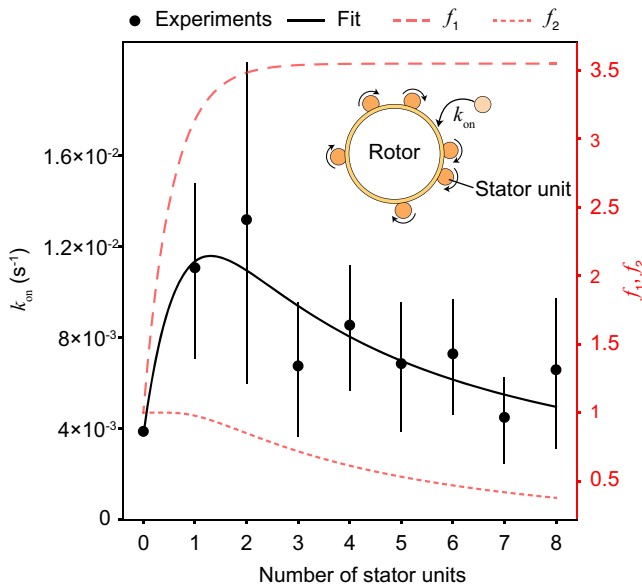

**Fig. 4 | Binding kinetics depend on the number of previously bound units.** The binding rate $k_{on}$ for an individual stator unit at different numbers of bound stator units ($N$). The black curve was obtained from the simple model $k_{on}(N) = k_{on}(0)f_1(N)$ $f_2(N)$, where $k_{on}(0) = 0.0037\,s^{-1}$ is the on rate for $N = 0$ and the two functions, $f_1$ and $f_2$, are given by $f_1(N) = (1 + A(1 - \exp[-BN]))$ and $f_2(N) = 1 - \exp[-C/N]$. The parameters $A$, $B$, and $C$ are determined from fitting the experimental data: $A = 0.23$, $B = 1.8$, $C = 3.8$. The data presented are means and the error bars represent standard deviation. The sample size for 0, 1, 2,... 8 stator units is 74, 63, 75, 70, 77, 81, 80, 83, and 74, respectively.

model and the experimental data offers $f_1$ and $f_2$ as a starting point for specific quantitative hypotheses for how the binding process is affected by motor rotation.

## Discussion

In this study, we combined quantitative measurements of the remodeling dynamics of single motors and first-passage-time analysis of the statistics of dwell times. We also developed and tested a minimal four-state model that quantitatively captures multiple experimental observations of the remodeling dynamics of the motor. A short-lived unbound state (H), proposed in a previous study of the steady-state motor dynamics[29], is confirmed in our analysis of the transient dynamics of motor remodeling. This H-state elucidates the existence of very short time scales in the distribution of dwell times, but it is not essential for mechano-adaptation. More importantly, our study reveals the existence of multiple states for a stator unit bound to the motor, each with a different unbinding rate. This multiplicity of bound states is what confers mechano-adaptation to the motor.

In all sensory adaptation systems, a key common feature is the existence of multiple internal states that can be modulated in response to changes in external stimuli[16]. For example, bacterial chemoreceptors have multiple methylation sites that allow chemoreceptors to adapt to changes in ligand concentrations by adjusting their methylation levels[34,35]. The multiple bound states of the stator unit identified here serve as the internal adjustable states necessary for adaptation of the bacterial flagellar motor to changes in external mechanical signals. The transitions between these bound states introduce additional timescales in the transient dynamics of motor remodeling, which allows us both to explain the experimental results and determine the transition rates between them.

As noted in the introduction, another recent study (performed concurrently with our work) proposed a model in which the bound

state is split into two states with different unbinding rates[28]. By analyzing population-averaged kinetics of stator remodeling, Perez-Carrasco et al. showed that a three-state model (also termed a 'two-state catch-bond model') can explain the asymmetry observed in the timescales of relaxation to steady state from either a large or small number of stator units[28]. However, these authors found that a previously proposed two-state (bound, unbound) model, which included a speed-dependent on rate[22], could also fit their data. While a careful Bayesian analysis did not favor the three-state model (which has more parameters) from the two-state model with a speed-dependent on rate, the three-state (loosely bound, tightly bound, unbound) model provided a better fit to the experimental observations. Therefore, the work by Perez-Carrasco et al. also provides a strong evidence for the presence of multiple bound states in the assembly of stator units in the flagellar motor.

Here, by analyzing the statistics of the dwell times of single motors, we conclusively show that a two-state model is incompatible with the experimental observations. We go a step further by including the short-lived hidden state in our model and confirming its presence in our experimental data. These advances are primarily enabled by the detailed analysis of single binding and unbinding events using first-passage-time methods, rather than by analyzing population averages. The convergence between our study and that of Perez-Carrasco et al., which investigated different aspects of mechanosensitive stator remodeling (dwell-time statistics vs. relaxation-time asymmetry), makes the model with multiple bound states a strong candidate for future theoretical and experimental work in this field.

The emerging picture for mechano-adaptive remodeling of the bacterial flagellar motor is that the torque generated by bound stator units controls their off rate by modulating the transition rates between the T-state (with a low or zero off rate) and the L-state (with a high off rate $k_{off,l}$), and thus the equilibrium between the two bound states. In particular, the ratio $\frac{k_l}{k_t}$ and consequently the effective off rate decrease with the torque $\Gamma$. As a result, a higher load leads to a higher $\Gamma$, which increases the number of bound stator units. This feedback mechanism, mediated by torque in the case of flagellar motor, is another key common feature of all adaptive systems[16].

Quantitatively, we find that the motor can tune its effective off rate over a wide range (>100 fold) from a very low rate ($\leq 0.0005\,s^{-1}$) at high torque (near stall) to a much higher rate (~0.057 $s^{-1}$) at low torque. Thus, the motor can adapt to changes in the mechanical load by adjusting the number of stator units over a range of $N_{tot}$ from 0 to 11. In our experiments, the motor drove the rotation of tethered cells and therefore operated close to stall. As a result, our data can only be used to determine the values of $k_t$ and $k_l$ at high torque (near stall). In the future, it will be interesting to use the modeling and analysis framework developed here to analyze data from experiments performed at different loads—for example, by using flagella labeled with beads of different sizes—to dissect the dependence of $k_l$ and $k_t$ on torque and to quantify the torque-mediated feedback mechanism.

As first pointed out by Nord et al.[23], the large reduction in the unbinding (off) rate of stator units with an increase in torque is reminiscent of the general catch-bond phenomenon, in which the lifetime of a receptor-ligand bond increases with tensile force applied to the bond[36]. In the case of stator units, however, the molecular mechanism causing the differences in off rates for different bound states remains unclear. In Fig. 2, we suggest a scenario in which the off rate may depend on the physical location (relative to its tethering position) of a bound stator unit, which is affected by the torque it generates. Another possibility is that the binding strength is increased by the torque through mechanically induced allostery in which torque production (and the resultant force) may propagate through the stator unit and cause conformational changes in the peptidoglycan binding domain of the stator unit. Indeed, these are only two of the several possible

scenarios that can lead to the catch-bond phenomenon[36]. Currently, there is no evidence to rule out either of these scenarios, both of which can be described by our model with two bound states (T and L). However, the presence of two bound states makes the binding of the stator unit to peptidoglycan analogous to the catch-bond behavior of FimH-mannose[37,38], kinetochore-microtubule[39], and vinculin-F-actin interactions[40], all of which also consist of two bound states. These contrast with the selectin-ligand[41] and myosin-actin[42] interactions, which exhibit a single time scale in their dwell time distributions.

What sets the relatively slow timescale of transition from the L state to the T state, as seen in a relatively small value of $k_t = 0.13$ s$^{-1}$? As we speculated before, the L and T states might correspond to different physical displacements of a bound stator unit. It is possible that such displacement of a bound unit is hindered by a large energy barrier in either the conformation space or the physical space that leads to a slow transition rate. Another intriguing possibility is that the rate $k_t$ is related to the recruitment of protein FliL, which has recently been shown (in *Borrelia burgdorferi* as well as in *Helicobacter pylori*) to form a ring-like structure around each bound stator unit[43,44]. In *B. burgdorferi*, FliL forms a partial ring before the binding of a stator unit, and oligomerizes into a full ring after a stator unit binds. This is believed to stabilize the bound stator unit in its active, torque-generating state. It is therefore possible that the L and T states respectively correspond to partial and full FliL ring around a bound stator unit. However, given that $k_t$ is a function of torque, how FliL oligomerization might be affected by torque is unclear. These ideas can be tested by measuring $k_t$ at different torque levels and at different expression levels of FliL. We leave these experiments to future studies.

Our study also reveals several interesting features of the dependence of the on rate on the number of stator units. Previously, Wadhwa et al. suggested a model in which the on rate decreases with motor rotation speed, and therefore with an increase in the number of stator units[22]. In contrast, recent work by Ito et al. found that the on rate increases with motor rotation speed, specifically when going from $\omega = 0$ to $\omega > 0$[33]. Our data reveal that the actual behavior is a combination of the two. The observed on rate has a non-monotonic dependence on the number of stator units: the on rate for $N = 0$ is smaller than that for $N = 1$ or 2, where it peaks. At $N > 2$, the on rate decreases. Indeed, a phenomenological model that combines features of these two competing effects successfully captures the observed trend. The non-monotonic dependence of the on rate on the number of bound stator units suggests that the on process may contain two (or more) steps with opposing dependence on the motor rotation speed. In principle, the on rate could depend on the number of bound stator units (via a possible cooperative binding effect) as well as on the motor speed $\omega$. However, in our current experiments at high load, $\omega$ is linearly proportional to $N$, which makes it impossible to separate the dependence on $N$ and $\omega$. Future experiments that systematically measure the on rate under different loads are needed to understand the on process and its dependence on $N$ and $\omega$.

A model with multiple bound states with mechanically regulated transition rates, such as the one proposed here for the bacterial stator units, also provides a possible mechanism for downstream signaling of mechanical signals. It is conceivable that, apart from regulating the binding strength of proteins, force could also alter their biochemical interactions with downstream signaling molecules. This could explain the putative role of the bacterial flagellar motor, and stator units in particular, in surface sensing during biofilm formation and differentiation of swarmer cells[45–48].

Overall, this work demonstrates the power of combining quantitative data from single molecule experiments with detailed stochastic analysis (e.g., first-passage-time analysis) to decipher the underlying mechanisms in biological systems without requiring that all the molecular details be known. Similar approaches should be applicable to studying other stochastic dynamic processes in biology, especially those that have to do with self-assembly of multi-protein complexes and their regulation by intrinsic or extrinsic signals.

## Methods

### Bacterial strains and cultures

We used *Escherichia coli* strain KAF95 (alias HCB986; a derivative of AW405) for all experiments. This strain is deleted for the chemotaxis response regulator CheY. Consequently, the cells of this strain rotate their flagellar motors exclusively counterclockwise. Additionally, this strain is deleted for the WT flagellin gene FliC and transformed with the plasmid pFD313, which expresses sticky FliC, resulting in filaments that readily attach to a variety of surfaces. Cells grown at 33 °C to OD$_{600}$ = 0.5–0.6 were washed and resuspended in TES buffer (20 mM TES, 0.1 mM EDTA, pH = 7.5). The cells were then sheared by passing through a 20 cm long piece of polyethylene tubing (inner diameter 0.58 mm) 60 times. The cells were then washed again to remove the sheared filaments and re-suspended in TES buffer.

### Electrorotation experiments

The electrorotation apparatus has been described before[22,49]. Cells were introduced into a custom-built flow cell that consisted of a circular sapphire window on one side and a circular glass cover slip on the other side. Cells readily tethered to sapphire via a short flagellar stub, which resulted in rotation of the cell body around the point of tether. The flow cell also contained four tungsten micro-electrodes whose tips were located at a short distance from the sapphire surface. The electrodes were driven in quadrature at 2.25 MHz to apply a rotating electric field on the tethered cells. This field caused an external torque on the cell body in the same direction as the torque applied by the flagellar motor. The strength of the external torque could be tuned by changing the amplitude of the rotating electric field. The temperature of the sapphire window was held constant at 20 °C by a circular Peltier element driven by a proportional controller. The flow cell and electrode assembly was fixed on the 20× objective of a phase contrast microscope. Rotating cells were imaged at 50 or 100 frames per second using a high-speed sCMOS camera (Edge 5.5; PCO-Tech). We selected such data from 58 different motors for further analysis described below. Data collection was automated using a program written in LabView 2017 SP-1.

### Data analysis and step fitting

Data analysis was performed using custom-written codes in Phython 3.6.4. We measured the angular displacement of the cell body between consecutive frames and multiplied it with the imaging frame rate, followed by filtering with a median filter of order 15. This provided the rotation speed of the motor as a function of time (gray line in Fig. 1). We then proceeded to fit steps to the rotation speed to extract the number of active stator units as a function of time. Of all the traces, we selected only those for further analysis in which the rotation speed was below 1 Hz at the end of the electrorotation phase, so that at most a single stator unit was bound to the motor at the beginning of the high load phase.

Our approach to step-fitting consisted of partitioning the total time in smaller time intervals in such a way that minimized the point-by-point distance between the fitted steps and the original trace. We called $\omega(t_i)$ the value of the rotation speed at time instant $t_i$ from the original data and, given that the time increment $\delta t$ was constant for every step, we had $t_i = i * \delta t$. We called $t_n$ the total time of a given experiment, where $n$ was the total number of measurements per trace. We needed to find an instant $t_l$ at which a step occured. The point of the partition (the index $l$) was chosen in such a way that the updated

step function was as close as possible to the original data. More formally, we defined the averages

$$\langle\omega\rangle_{0,l} = \frac{1}{l}\sum_{k=0}^{l}\omega(t_k), \tag{6}$$

and

$$\langle\omega\rangle_{l,n} = \frac{1}{n-l}\sum_{k=l}^{n}\omega(t_k), \tag{7}$$

where $\langle\omega\rangle_{0,l}$ was the average of the original data in the time interval between $t_0 = 0$ and $t_l$ and $\langle\omega\rangle_{l,n}$ was the average of the data in the interval between $t_l$ and the final instant $t_n$. Then, $l$ was chosen in such a way that the residual $\Delta$, defined as sum of the squares of the difference between the fitted step and the original data at each time instant:

$$\Delta = \sum_{i=0}^{l}\left(f(t_i) - \langle\omega\rangle_{0,l}\right)^2 + \sum_{i=l}^{n}\left(\omega(t_i) - \langle\omega\rangle_{l,n}\right)^2, \tag{8}$$

was minimal. The partition was then iterated several times. The formula for $\Delta$ after $K$ iterations was

$$\Delta = \sum_{i=0}^{K-1}\left[\sum_{k=l_i}^{l_{i+1}}\left(\omega(t_k) - \langle\omega\rangle_{l_i,l_{i+1}}\right)^2\right], \tag{9}$$

where $l_i$ was the index for the instant of at which the $i$-th step in the fitted function occured and the indices were defined so that $l_i < l_{i+1}$ for all values of $i$ ($i = 0, 1, \ldots, K-1$).

The choice of the total number of iterations $K$ was important in order to avoid over-fitting. To determine when to interrupt the loop, at each iteration we calculated the average $\tilde{\Delta} = \langle\Delta\rangle_\alpha$ that was obtained by partitioning at randomly generated instants $t_\alpha$ rather than the specific instants that minimize $\Delta$ to $\Delta_{min}$. Iteration was stopped as soon as $\Delta_{min}/\tilde{\Delta} > 0.995$, i.e., when compared to an equal number of randomly placed steps, an additional computed step did not improve the fit by more than 0.5%.

The process described above resulted in a fitted step function $g$ for a given time $t_u$, where $l_i < u < l_{i+1}$:

$$g(t_u) = \langle\omega\rangle_{l_i,l_{i+1}}. \tag{10}$$

At this stage, we needed to associate the discrete values of $g$ to a specific number of stator units. To do so, we determined whether slightly different values of $g$ during different time intervals corresponded to the same number of stator units. This was done with a sorting method. First, we sorted all the values of $g$ in ascending order. Then, if two consecutive values of $g$ (levels) differed by less than 0.75 Hz, we assigned to them a new level given by the weighted average of their current levels. Then, we associated a number of stator units $N$ to each level starting with $N = 0$ for the level with a value close to 0 Hz and we added a stator unit for each subsequent level.

### Reporting summary
Further information on research design is available in the Nature Research Reporting Summary linked to this article.

## Data availability
The raw data used in all analyses presented here are available from the GitHub repository https://github.com/navishwadhwa/multi-state-remodeling. Source data for Figs. 1c, 3, and 4 are provided with this paper. Source data are provided with this paper.

## Code availability
All codes used are available from the GitHub repository https://github.com/navishwadhwa/multi-state-remodeling.

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

## Acknowledgements
We are grateful to Prof. Mike Manson for his critical reading of the manuscript. This work was supported by the National Institutes of Health under Award Numbers K99GM134124 (to N.W.) and R35GM131734 (to Y.T.). This publication is dedicated to the memory of our co-author, Prof. Howard C. Berg, who was a world-renowned leader in the studies of bacterial motility and chemotaxis. Sadly, Howard passed away during the final phases of this work.

## Author contributions
N.W. and A.S. contributed equally to this work. N.W. jointly conceived the study with Y.T., built the electrorotation apparatus together with H.C.B., conducted the initial data analysis, and co-wrote the manuscript; A.S. performed all subsequent data analysis, model fitting, simulations, and co-wrote the manuscript; Y.T. developed the analytic model, supervised the data analysis, and co-wrote the manuscript.

## Competing interests
The authors declare no competing interests.
