## [Peer Review File · Nature Communications]

REVIEWER COMMENTS

Reviewer #1 (Remarks to the Author):

This paper describes efforts to dissect the mechanism by which stator units in the flagellar motor bind more tightly when the load is increased. The problem is of some interest as it is a relatively little studied kind of biological adaptation. The data are more extensive than in previous studies and the first-passage-time analysis is in principal a useful improvement over previous analyses. I have a few concerns with the paper, described below.

1. An excerpt from the paper:

"It [a simpler model] predicts that the dwell time of the stator complex at any given level will be exponentially distributed (see SI). To test this prediction, we plotted the distribution of dwell times at all stator unit numbers"

In other words, the data plotted in Fig. 1c in support of the more-complex model is for all levels, whereas the statement regarding the single-exponential behavior expected in a simple model pertains only to each level individually.

The plots will obviously be more noisy if data for each level are plotted separately, but it seems to me that such level-by-level analysis is needed to really make the point (I realize that statements about individual levels are provided further into the paper, and will address that below). As a compromise, what if data are binned into sets of 3 consecutive levels, or something along those lines? Is the non-simple-exponential character decreased? If so, then I feel that it would be necessary to present that, so the reader has a sense of the degree (if any) to which the non-simple character of the plots reflects that fact that all levels have been grouped together.

2. Related: What do simulations look like if we assume that the on-rate has only the N_t - N dependence, and off-rates have a single time-independent value? Because of the N_t - N dependence, it seems that there will be some non-single-exponential character in the dwell times, though I'm not sure. How great is the deviation from simple behavior in such a model?

3. In the later parts of the paper, some analysis is carried through for individual levels, as is necessary, and it is stated that the behavior even within a single level is not simple exponential (based on a variance larger than expected in a simple exponential model). First, since it's made clear here that level-by-level analysis is what's needed, the plotting of all levels together in Fig. 1c seems all the more puzzling. Second, and more importantly, since it is the variance in dwell times that is used to argue for non-simple behavior, it seems that other possible sources of this variance should be considered. Though the present experiments monitor the behavior of individual cells, the data on any given level (I believe) represent data obtained from more than one cell. Even in the simplest model, the observed on-rate for stators will be the product of a bimolecular rate constant and an effective concentration of the stators in the membrane pool. This pool concentration might vary from cell to cell; if it varies by a factor of two or more, couldn't that provide an alternative explanation for the variance in dwell times?

4. The rate of the transition to the tight state (in the model presented) is 0.13 per second. If this just reflects the increase in force applied to the stator, isn't it unexpectedly slow? What sort of event is envisioned to account for this slow rate, given that a stator, once functioning, would be expected to achieve its equilibrium torque more quickly than this?

Reviewer #2 (Remarks to the Author):

In the manuscript the authors explore distributions stator occupancy survival states showing that a multistate model results in a better fit of the data. While the paper is overall interesting, a more precise statistical methodology is necessary to back the claims of the paper. In addition, a recent study (Perez-Carrasco et al 2021 in the paper) proposed a similar model of the multi-state dynamic in a preprint uploaded last summer. While the results of both manuscripts are compatible, they take novelty out from the current paper. At least a more comprehensive introduction that makes a reference to this previous paper where the multi-state model is introduced would be required. Please find enclosed more specific comments:

1) One of the factors limiting the interpretability of the results is the amount of data. While the authors make several fittings along the text to their model, adequate statistical analysis is necessary to back their claims. This is the case for instance in the detection of 3 different timescales in Fig.1c. The number and location of the scales should be obtained using an objective statistical method.

2) Similarly the fittings to the experimental data was insufficient. The authors have theoretical and experimental distributions of dwell times. The parameters that fit these distributions should be obtained by comparing both probability distributions using maximum likelihood estimation. Otherwise, using arbitrary summary statistics with the same weight could bias the inference.

3) If I understood correctly, the values of k_{on} reported in Fig.4 as experimental are obtained using the experimental dwell times and the experimental fraction f_+ . Were those the same values used to fit the curves in Fig.4 (labelled as experiments), and used in Fig.3? In the description of the supplementary "The model parameters are obtained with an optimization algorithm" it looks like $k_+(N)$ is fitted to the data as part of the loss function.

4) It was not clear to me if the results of Fig.3 correspond to the full model or to the limit ($c_1=c$), or the fit including c_1, c_2 , and c_3 . At the end of page 7, the authors only mention c . On the other hand, in the SI they mention the values c_1, c_2, c_3 .

5) I failed to understand the relevance of equation 5. While it provides a heuristic functional form for $k_{on}(N)$, I do not think that it reveals a mechanism. It shows that the experimental non-monotonous k_{on} can be written as a product of two ad hoc monotonous functions. Without a direction on how $f_1(N)$ and $f_2(N)$ provide a testable mechanism for this composition, the claims are purely speculative. I agree with the discussion that the authors provide on opposite effects of occupancy/speed and recruitment rates, but I don't think it requires to invoke a mathematical description (or a fitting to it) to support that argument.

6) As I mentioned before, the authors make reference in the discussion to a previous manuscript that already presents a multi-state model. Actually, since the hidden-state dwells were removed from the analysis, both models are equivalent. Not only the authors didn't mention this model in the introduction but also provide an unsatisfactory reference in the discussion. The authors say that in [Perez-Carrasco et al] the authors could not distinguish between the previous ON-OFF model and a multistate model. This is so because in [Perez-Carrasco et al] the authors used a Bayesian method that takes into account the dimensionality of the parameter space. If the analysis is performed just in terms of a loss function (as it is done in the current manuscript), then the fitting of the multi-state model in [Perez-Carrasco et al] is much better than the original ON-OFF model, obtaining the same results as in the current paper. This is not surprising since both papers base their argument in the absence of a Markovian exponential behaviour.

Reviewer #3 (Remarks to the Author):

The authors performed a detailed experimental study of the bacterial flagellar motor in *E. Coli*. The motor is composed of a central rotor and torque is generated by stator units, which are known to bind and unbind from the rotor in a torque-sensitive manner. While models have been proposed and analyzed for these binding dynamics that capture the average behavior well, these models also make specific predictions for the stochastic behavior -- in particular the dwell time statistics of the stators -- which has yet to be experimentally tested in detail.

Using a rotating electric field, the authors spun a bacteria tethered to a surface, effectively decreasing the load, causing the stators to unbind. Once the electric field is suddenly turned off, the authors monitored the rotation speed at the increased load. Discrete jumps in speed corresponding to the

addition of individual stator units were monitored. The dwell times between additions were measured and analyzed.

The dwell times displayed a behavior inconsistent with a single exponential, which suggests multiple internal stator binding states. The authors proposed a model with 4 stator states, which fit the experimental data fairly well. Estimates for the rates of transitions were obtained by numerically fitting the lower order dwell time statistics – mean and variance – to the model predictions. Estimates were reasonable in light of previous studies. Interestingly, they also observed an on-binding rate that depended on the number of stators present, which they could fit with a phenomenological model.

The paper is clear and well-written. Bacterial flagellar motors are known not to be static, but dynamically remodeling in a torque-dependent manner. This study on-the-level of single motors is very suggestive that an accurate model of the kinetics of stator binding requires more than the two states (or two+hidden states) as previously suggested. An accurate understanding of the motors remodeling and mechanosensitive will require accurate models of the kinetics. In this respect, this is a timely and important study sure to be stimulating to the community. I therefore offer my recommendation for publication.

Minor suggested revisions:

- 1) In the SI, just before (S2), could the authors justify their use of the “long time approximation” here.
- 2) In the SI, just before (S10) the authors set $k_{\text{off},t}=0$. This confused me at first, since in the main text, it is not zero, merely very small. A suggested edit, would be to clarify that by setting $k_{\text{off},t} = 0$, it is the first step in an approximation where it is small, and not identically zero to align better with the main text.

Reviewer 1

This paper describes efforts to dissect the mechanism by which stator units in the flagellar motor bind more tightly when the load is increased. The problem is of some interest as it is a relatively little studied kind of biological adaptation. The data are more extensive than in previous studies and the first-passage-time analysis is in principal a useful improvement over previous analyses. I have a few concerns with the paper, described below.

We thank the reviewer for providing many insightful comments on our manuscript. Below we provide a response to the specific comments and outline the changes we have made in the manuscript to address these comments.

1. An excerpt from the paper: “It [a simpler model] predicts that the dwell time of the stator complex at any given level will be exponentially distributed (see SI). To test this prediction, we plotted the distribution of dwell times at all stator unit numbers” In other words, the data plotted in Fig. 1c in support of the more-complex model is for all levels, whereas the statement regarding the single-exponential behavior expected in a simple model pertains only to each level individually. The plots will obviously be more noisy if data for each level are plotted separately, but it seems to me that such level-by-level analysis is needed to really make the point (I realize that statements about individual levels are provided further into the paper, and will address that below). As a compromise, what if data are binned into sets of 3 consecutive levels, or something along those lines? Is the non-simple-exponential character decreased? If so, then I feel that it would be necessary to present that, so the reader has a sense of the degree (if any) to which the non-simple character of the plots reflects that fact that all levels have been grouped together.

Figure 1. Semi-log plot of the dwell time distributions for different values of $N = 0, 1, 2, \dots, 7$ (the number of bound stator units).

The distribution of dwell times for individual levels is only one of the many aspects of the simple (two-state) model that do not agree with the experimental data. Other such aspects include the range of change in mean dwell times, the normalized variance of dwell times for each level, the differences between the ‘+’

and ‘-’ event dwell times, as well as the differences between the overall distribution in experiments vs simulations of the two state model (see below).

We have significantly expanded the section entitled “The dwell time statistics in two-state model with only one bound state” in the SI where we discuss these disagreements between the two-state model and our experiments in more detail. We have also revised the passage quoted above in the main text. Here we chose to show the distribution of dwell times during the entire recovery process over different values of N to reduce noise and enhance interpretability. To further aid the reader in interpreting the non-simple character of this distribution, we have modified Fig. 1c to include a direct comparison with the simulation results of a two-state model (see our response to the next comment and the revised Fig. 1c). We believe that this comparison effectively highlights the disagreement between the two-state model and the experimental data, which motivates the development of a model with additional bound state(s).

To respond to the referee’s comment directly, we plot the dwell time distributions for each individual level in Fig. 1 of this letter. As expected, the plots for individual levels are noisier than in the case of a distribution of dwell times for all levels taken together (Fig. 1c). However, it is still possible to appreciate that the long time tails of the distribution become progressively more prominent as N increases, which can not be described by an exponential distribution with a single time scale. This is also supported by the fact that the normalized variance V is larger than 1 for all values of N , as shown in Fig. 3c in the main text.

In the revised SI, we have now included these plots of dwell time distribution for individual levels in the section on the two-state model, in which we explained in quantitative details why these observed dwell time distributions for different values of N are inconsistent with the two-state model even when the dependence of k_+ on N is included.

2. Related: What do simulations look like if we assume that the on-rate has only the Nt - N dependence, and off-rates have a single time-independent value? Because of the Nt - N dependence, it seems that there will be some non-single-exponential character in the dwell times, though I’m not sure. How great is the deviation from simple behavior in such a model?

Thank you for this suggestion. We ran simulations for a simple two-state model in which the binding-unbinding dynamics for each stator unit with the off-rate having a single time-independent value k_{off} and an on-rate that is proportional to $(N_{\text{tot}} - N)$ as suggested by the referee (we also took out the **H** state as it is not relevant for the discussion here). We sampled the same number of traces as in the experiments, each running for a total time the same as that in our experiments.

In Fig. 2 of this letter, we show the simulation results for two different values of k_{off} : $k_{\text{off}} = 0.0005 \text{ s}^{-1}$ (blue dots), and $k_{\text{off}} = 0.05 \text{ s}^{-1}$ (red dots) to cover the full range of possible values of k_{off} . We also plotted the dwell time distribution from experiments (triangles) for comparison. The qualitative difference between the two-state model results and the experiments is clear: while the change of time scale in experiments can be clearly seen at around $\tau \sim 50\text{s}$, where the slope of $\ln(P)$ versus τ dependence changes significantly, such change in the two-state model is much weaker. Quantitatively, the two-state model with a larger value of $k_{\text{off}}=0.05 \text{ s}^{-1}$ agrees with the experiments for smaller values of $\tau \leq 50\text{s}$, but underestimates the long time scale events ($\tau > 50\text{s}$). On the other hand, for a smaller value of $k_{\text{off}}=0.0005 \text{ s}^{-1}$, the model captures some of the long time scale events, but it disagrees with the experiments for the short time events.

In the revised manuscript, we have replaced the previous Fig. 1c with this comparison figure (Fig. 2 here) to demonstrate the disagreement between the two-state model and our experiments.

3. In the later parts of the paper, some analysis is carried through for individual levels, as is necessary, and it is stated that the behavior even within a single level is not simple exponential (based on a variance larger

Figure 2. Distribution of dwell times for the two-state model (green triangles and purple discs) with two different values of the dissociation constant k_{off} , in comparison with the distribution from experiments (red squares).

than expected in a simple exponential model). First, since it's made clear here that level-by-level analysis is what's needed, the plotting of all levels together in Fig. 1c seems all the more puzzling. Second, and more importantly, since it is the variance in dwell times that is used to argue for non-simple behavior, it seems that other possible sources of this variance should be considered. Though the present experiments monitor the behavior of individual cells, the data on any given level (I believe) represent data obtained from more than one cell. Even in the simplest model, the observed on-rate for stators will be the product of a bimolecular rate constant and an effective concentration of the stators in the membrane pool. This pool concentration might vary from cell to cell; if it varies by a factor of two or more, couldn't that provide an alternative explanation for the variance in dwell times?

Thank you for this important question regarding the possible effects of cell-to-cell variability on the statistics of dwell times and in particular on the variability in dwell times for a given stator unit number N . The argument is that if there is a large enough variability in rates, especially k_+ due to cell-to-cell variation in the number of freely diffusing stator units, then even within a 2-state model such cell-to-cell variability could potentially account for the large observed values of $V \equiv \frac{\langle \tau^2 \rangle}{\langle \tau \rangle^2} - 1$, where τ is the dwell time. We agree that cell-to-cell variability would increase V . However, as we show below, the large value of V (≈ 2) observed in our experiments can not be explained by cell-to-cell variability because that would require an unrealistically high level of variability in the kinetic rates in a highly correlated fashion.

In a two-state model, the statistics of dwell time are determined by the total rate $k = k_+ + k_-$, which is the sum of the on-rate (k_+) and the off-rate (k_-). For a given k , τ follows an exponential distribution: $P(\tau|k) = k \exp(-k\tau)$ with its first and second moments given by $\langle \tau \rangle = k^{-1}$ and $\langle \tau^2 \rangle = 2k^{-2}$, respectively. Now, let us assume that the cell-to-cell variation in k can be described by a distribution $q(k)$ for k . We can

then express the normalized variance V as:

$$V \equiv \frac{\langle \tau^2 \rangle}{\langle \tau \rangle^2} - 1 = \frac{\int 2k^{-2}q(k)dk}{(\int k^{-1}q(k)dk)^2} - 1. \quad (1)$$

For simplicity, let us assume that k has a broad uniform distribution between k_1 and k_2 , i.e., $q(k) = (k_2 - k_1)^{-1}$ when $k_1 \leq k \leq k_2$ and 0 otherwise. This distribution is characterized by a maximum fold change $f = k_2/k_1 \geq 1$. From Equation 1, one can show that V depends on the fold change f as:

$$V = \frac{2(f-1)^2}{f(\ln f)^2} - 1, \quad (2)$$

which has a rather weak dependence on f (see Fig. 3 below). When $f \rightarrow 1$, i.e., there is no variation in k , $V \rightarrow 1$. V increases with f , but slowly. For V to reach the observed value of ≈ 2 , f has to be as large as 10. For a more reasonable value of $f = 4$, we have $V = 1.34$, which is far below the observed value of V .

Figure 3. The dependence of the normalized variance V on the maximum fold change f . A large value of $f \approx 10$ is required in order to have $V = 2$ (the dashed line).

Another important issue is that since $k = k_+ + k_-$, a large fold change in k requires both k_+ and k_- to have the same (or similar) fold change in a correlated way. If the variations in k_+ and k_- are uncorrelated, which we believe is the case here, the variation in k measured by fold change will be dominated by the rate that has the higher average value. At higher value of N , k_- is comparable to and even larger than k_+ , in which case the variation in k could be dominated by variation in k_- . However, within the 2-state model, there is no obvious reason for k_- to have a large variability. For example, if we assume that k_- has a relatively small variation and the averages of k_+ and k_- are the same (\bar{k}), then even a highly variable distribution for k_+ , e.g., a bimodal distribution $P(k_+) = [\delta(k_+) + \delta(k_+ - 2\bar{k})]/2$ with a maximum fold

change for k_+ , $\frac{k_{+,max}}{k_{+,min}} \rightarrow \infty$ would only lead to a modest fold change in k : $f \approx 3$, which is far below what is needed to achieve $V \approx 2$ within the 2-state model.

In the revised SI, we now include these discussions regarding effects of the cell-to-cell variability and the reasons why they can not be the main cause for the observed large variability in the dwell time statistics.

4. The rate of the transition to the tight state (in the model presented) is 0.13 per second. If this just reflects the increase in force applied to the stator, isn't it unexpectedly slow? What sort of event is envisioned to account for this slow rate, given that a stator, once functioning, would be expected to achieve its equilibrium torque more quickly than this?

We do not believe that the timescale of transition from the **L** state to the **T** state, as determined by k_t , is related to the timescale for torque generation, which happens immediately upon the binding of a stator unit at the periphery of the rotor. In fact, there is no difference between the **L** state and the **T** state in terms of torque generation. Furthermore, the transition rates must be considered relative to each other. k_t is actually larger than the other rates, in particular the off rate from the **L** state, $k_{off,l}$, which is found to be $\sim 0.057 \text{ s}^{-1}$. k_t can also not be too large. If $k_t \gg k_{off,l}$, e.g., if $k_t = 100k_{off,l} = 5.7 \text{ s}^{-1}$, then once bound, with a high probability ($> 99\%$) a bound stator unit would go from the **L** state to the **T** state and stay there since k_l is very small. In that case, there would be almost no unbinding events with a tiny rate $\sim 10^{-2}k_{off,l} = 5.7 \times 10^{-4} \text{ s}^{-1}$, which is certainly not what we observed.

It is possible that the **L** state and **T** state are separated by a large energy barrier in either the conformation space or the physical space that leads to a slow transition rate. We do not fully know the molecular details of the **L** and **T** states, nor the precise mechanism underlying the transition from one state to the other, so it is difficult to interpret the magnitude of individual rates. One intriguing possibility is that the rate k_t is related to the recruitment of protein FliL, which has recently been shown (in *Borrelia burgdorferi* as well as in *Helicobacter pylori*) to form a ring-like structure around each bound stator unit^{1,2}. Particularly in *B. burgdorferi*, FliL was shown to form a partial ring at the periphery of the motor before the binding of a stator unit. After a stator unit binds, FliL oligomerizes into a full ring that is believed to stabilize the bound stator unit in its active, torque-generating state. It is possible that the **L** and **T** states respectively correspond to partial and full FliL ring around a bound stator unit. If that is the case, then the rate of transition from the **L** state to the **T** state (k_t) is determined by the oligomerization rate of FliL. However, given that k_t is a function of torque, how FliL oligomerization might be affected by torque remains unclear. These ideas can be tested by measuring k_t at different torque levels and at different expression levels of FliL. We leave these experiments to future studies.

*We have revised the text to clarify the differences and similarities between the **L** and **T** states. In addition, we now include the possible role of FliL in stator remodeling in the Discussion section.*

Reviewer 2

In the manuscript the authors explore distributions stator occupancy survival states showing that a multistate model results in a better fit of the data. While the paper is overall interesting, a more precise statistical methodology is necessary to back the claims of the paper. In addition, a recent study (Perez-Carrasco et al 2021 in the paper) proposed a similar model of the multi-state dynamic in a preprint uploaded last summer. While the results of both manuscripts are compatible, they take novelty out from the current paper. At least a more comprehensive introduction that makes a reference to this previous paper where the multi-state model is introduced would be required. Please find enclosed more specific comments:

We thank the reviewer for providing detailed feedback and constructive criticism for our manuscript. Below we address each of the specific comments and outline the changes we have made (in italic) in the manuscript to address these comments.

1. One of the factors limiting the interpretability of the results is the amount of data. While the authors make several fittings along the text to their model, adequate statistical analysis is necessary to back their claims. This is the case for instance in the detection of 3 different timescales in Fig. 1c. The number and location of the scales should be obtained using an objective statistical method.

We agree with the reviewers about the importance of adequate statistical analysis for noisy biological data. Indeed, guided by a mechanistic model, analyzing the statistics of the noisy motor remodeling data is exactly what we did in this study. The data reported in this study represent high quality single motor measurements of over 50 individual motors over 6 minutes each during their remodeling process. Of course, more data would be better, but we believe that we have made efficient use of the data presented here to determine the general nature of the motor remodeling process.

By showing three different timescales in Fig. 1c, we only intended to make the point that there are multiple additional timescales in the system. As we later show in the paper that there are actually a continuous spectrum of timescales due to the existence of the **T** and **L** bound states. *In the revised version of Fig. 1c, we now show a comparison between the experimentally obtained distribution of dwell times and simulations of two different scenarios of the two-state model. This obviates the need for timescale fits and motivates the development for a model with multiple bound states that introduce additional timescales to the distribution of dwell times.*

2. Similarly the fittings to the experimental data was insufficient. The authors have theoretical and experimental distributions of dwell times. The parameters that fit these distributions should be obtained by comparing both probability distributions using maximum likelihood estimation. Otherwise, using arbitrary summary statistics with the same weight could bias the inference.

The aim of our modeling approach is to better understand the key biophysical aspects of the system, as characterized by physically meaningful quantities such as the fraction of “+” transition events (f_+), the mean dwell time ($\langle\tau\rangle$), and the magnitude of variation in the dwell time as measured by V (the variance of the dwell time normalized by the square of its mean). We therefore used our model to fit these quantities and their dependence on N , all of which can be measured directly in our experiments. While these quantities can all be obtained from the distribution function $P_{\pm}(\tau|N)$, they are not arbitrary summary statistics of the distribution function. We chose these for their relevance in understanding the underlying motor dynamics. The fact that these quantities are the lowest moments of the distributions is also important

because higher order moments would have larger statistical noise from the experiments. Indeed, we did consider the experimental measurement error in our fitting by having the weight for each loss function term depend inversely on the experimental measurement error. *We have now revised our paper to clarify the reasons for fitting these biophysically meaningful properties of the dwell time distributions before and after Eq. 4 in the main text where we introduce f_{\pm} , $\langle\tau\rangle$, and V .*

Additionally, it is a standard practice to fit a model distribution to an observed distribution by first fitting its lowest moments. Especially when the number of parameters in the model is small (compared to the degree of freedom, as is the case here) and the data are noisy, fitting the lowest-order moments is a simple and robust approach to determine parameter values. Once the parameters are determined this way, they may be further refined by using the maximal likelihood estimation (MLE), which fit the full distribution that include in principle all the higher order moments of the distribution. Unless the model is inconsistent with the data, the “adjustment” of these parameters is typically small and they may not lead to better fitting of the physically meaningful properties of the system such as $\langle\tau\rangle$ and f_{+} .

We performed MLE for our problem, and indeed it does not change the results significantly. In particular, we performed the optimization again using a MLE loss function of the following form:

$$L_{MLE}(p, x) = \sum_N \sum_i D[P^{exp}(x, i dt, N), P^{th}(p, i dt, N)] + \sum_N (f_{+}^{exp}(x) - f_{+}^{th}(p))^2, \quad (3)$$

where x are the experimental data, p the fitting parameter, P^{exp} and P^{th} the dwell time distributions for a given number of stator units, D is a measure of the distance between the two functions and dt the bin size we chose to evaluate the distributions. Here, we used the mean squared difference for D . Other distance measures such as the log likelihood function (or the Kullback–Leibler divergence) lead to poor fitting of the distribution and $\langle\tau\rangle$ because these distance measures tend to overemphasize the outliers in the dwell time distribution at large τ where the noise in the measured distribution is high.

Figure 4. Comparison between experiments (red symbols and line) and model predictions (black symbols) with parameters obtained from MLE for (a) $\langle\tau\rangle$, (b) f_{+} , and (c) V .

We used the gradient descent method to minimize L_{MLE} and the parameters obtained are not significantly different from those we reported in the article, namely $c_1 = 0.39$ (0.3 ± 0.1 in the article), $c_2 = 0.14$ (0.12 ± 0.14 in the article), $c_3 = 0.012$ (0.06 ± 0.16 in the article), $\sigma_{+} = 0.09$ (0.19 ± 0.1 in the article). The predicted values for τ , f_{+} , and V with the adjusted parameters agree with the experiments within the experimental error bars. As shown in Fig. 4 here, the agreements for f_{+} and V with the parameters obtained by MLE are similar to those shown in Fig. 3b,c in the main text by using the original (un-adjusted) parameters obtained by minimizing the loss function given in Eq. S28 in the SI,

but the agreement for $\langle \tau \rangle$ is worse (see Fig. 3a in the main text). Therefore, at least for our problem, the MLE-based fitting method does not out-perform the moment-based method used in our paper.

We have now included a discussion about this MLE-based approach for parameter estimation and how it compares to the fitting method used in our study.

3. If I understood correctly, the values of k_{on} reported in Fig.4 as experimental are obtained using the experimental dwell times and the experimental fraction f_+ . Were those the same values used to fit the curves in Fig.4 (labelled as experiments), and used in Fig.3? In the description of the supplementary “The model parameters are obtained with an optimization algorithm” it looks like $k_+(N)$ is fitted to the data as part of the loss function.

This is correct. The values of experimental k_{on} in Fig. 4 are obtained from the experimental $\langle \tau \rangle$ and f_+ as reported in Fig. 3.

In our model fitting, we treat $k_+(N)$ as fitting parameters and use our model to fit the direct experimental observables $f_+(N)$ and $\langle \tau \rangle(N)$ (see our response to previous comment), both of which depend on $k_+(N)$. This can be seen by the following terms in the loss function:

$$\sum_{N=0}^9 \left[\alpha_1 (\langle \tau \rangle^{\text{exp}} - \langle \tau \rangle^{\text{mod}}(p, N))^2 + \alpha_2 (f_+^{\text{exp}} - f_+^{\text{mod}}(p, N))^2 \right],$$

which does not explicitly contain k_+ -dependent terms. However, given that $k_+ = f_+ / \langle \tau \rangle$, this loss function effectively forces the model k_+^{mod} to be close to the experimental value k_+^{exp} within the experimental error of f_+ and $\langle \tau \rangle$.

4. It was not clear to me if the results of Fig.3 correspond to the full model or to the limit ($c_1 = c$), or the fit including c_1 , c_2 , and c_3 . At the end of page 7, the authors only mention c . On the other hand, in the SI they mention the values c_1 , c_2 , c_3 .

Fig. 3 corresponds to the full model with c_1 , c_2 and c_3 . *We now state this explicitly in the SI where details of the model results and fitting in the paper are described.*

5. I failed to understand the relevance of equation 5. While it provides a heuristic functional form for $k_{\text{on}}(N)$, I do not think that it reveals a mechanism. It shows that the experimental non-monotonous k_{on} can be written as a product of two ad hoc monotonous functions. Without a direction on how $f_1(N)$ and $f_2(N)$ provide a testable mechanism for this composition, the claims are purely speculative. I agree with the discussion that the authors provide on opposite effects of occupancy/speed and recruitment rates, but I don't think it requires to invoke a mathematical description (or a fitting to it) to support that argument.

We agree that Eq. 5 does not reveal a mechanism. However, the two functions f_1 and f_2 are not arbitrary. f_2 is a simplified form of the speed-dependent on-rate first proposed in Wadhwa, Phillips and Berg (ref. 22 in the main text). It codifies the notion of a contact time (between an unbound stator unit and a subunit of the rotor) that is reduced due to the rotation of the rotor. f_1 is chosen as the simplest possible function that is consistent with the the recently published work by Ito and coworkers which suggests that the on rate is enhanced by rotation. Together, these two functions provide a quantitative basis for the underlying hypotheses for how the on-rate might depend on rotation speed.

To clarify these points, we have made the following changes in the text (on page 9, at the end of the results section): “While the functional form of $k_{\text{on}}(N)$ is speculative, the agreement between this model

and the experimental data offers f_1 and f_2 as a starting point for specific quantitative hypotheses for how the binding process is affected by motor rotation.”

6. As I mentioned before, the authors make reference in the discussion to a previous manuscript that already presents a multi-state model. Actually, since the hidden-state dwells were removed from the analysis, both models are equivalent. Not only the authors didn't mention this model in the introduction but also provide an unsatisfactory reference in the discussion. The authors say that in [Perez-Carrasco et al] the authors could not distinguish between the previous ON-OFF model and a multistate model. This is so because in [Perez-Carrasco et al] the authors used a Bayesian method that takes into account the dimensionality of the parameter space. If the analysis is performed just in terms of a loss function (as it is done in the current manuscript), then the fitting of the multi-state model in [Perez-Carrasco et al] is much better than the original ON-OFF model, obtaining the same results as in the current paper. This is not surprising since both papers base their argument in the absence of a Markovian exponential behaviour.

We regret that our discussion of the study by Perez-Carrasco *et al.* was found unsatisfactory. Our study was conducted concurrently with Perez-Carrasco *et al.* and we became aware of it when it was released as a preprint. For this reason, we originally chose to refer to Perez-Carrasco *et al.* in the discussion.

Following the reviewer's suggestion, we have revised both the introduction and the discussion sections, in an attempt to provide a more balanced view of work by Perez-Carrasco et al. In the Introduction (page 2), we have added: “Indeed, recent work by Perez-Carrasco and co-workers found that the asymmetry observed in the timescales of relaxation to steady state (in population-averaged data) from either a large or a small number of stator units could not be explained by a simple, two-state model. This gap could be best resolved by the introduction of a third, strongly bound state into the model.”

In the discussion: “As noted in the introduction, another recent study (performed concurrently with our work) proposed a model in which the bound state is split into two states with different unbinding rates. By analyzing population-averaged kinetics of stator remodeling, Perez-Carrasco et al. showed that a three-state model (also termed a ‘two-state catch-bond model’) can explain the asymmetry observed in the timescales of relaxation to steady state from either a large or small number of stator units. However, these authors found that a previously proposed two-state (bound, unbound) model, which included a speed-dependent on rate, could also fit their data. While a careful Bayesian analysis did not favor the three-state model (which has more parameters) from the two-state model with a speed-dependent on rate, the three-state (loosely bound, tightly bound, unbound) model provided a better fit to the experimental observations. Therefore, the work by Perez-Carrasco et al. also provides a strong evidence for the presence of multiple bound states in the assembly of stator units in the flagellar motor.”

Reviewer 3

The authors performed a detailed experimental study of the bacterial flagellar motor in *E. coli*. The motor is composed of a central rotor and torque is generated by stator units, which are known to bind and unbind from the rotor in a torque-sensitive manner. While models have been proposed and analyzed for these binding dynamics that capture the average behavior well, these models also make specific predictions for the stochastic behavior — in particular the dwell time statistics of the stators – which has yet to be experimentally tested in detail.

Using a rotating electric field, the authors spun a bacteria tethered to a surface, effectively decreasing the load, causing the stators to unbind. Once the electric field is suddenly turned off, the authors monitored the rotation speed at the increased load. Discrete jumps in speed corresponding to the addition of individual stator units were monitored. The dwell times between additions were measured and analyzed.

The dwell times displayed a behavior inconsistent with a single exponential, which suggests multiple internal stator binding states. The authors proposed a model with 4 stator states, which fit the experimental data fairly well. Estimates for the rates of transitions were obtained by numerically fitting the lower order dwell time statistics – mean and variance – to the model predictions. Estimates were reasonable in light of previous studies. Interestingly, they also observed an on-binding rate that depended on the number of stators present, which they could fit with a phenomenological model.

The paper is clear and well-written. Bacterial flagellar motors are known not to be static, but dynamically remodeling in a torque-dependent manner. This study on-the-level of single motors is very suggestive that an accurate model of the kinetics of stator binding requires more than the two states (or two+hidden states) as previously suggested. An accurate understanding of the motors remodeling and mechanosensitive will require accurate models of the kinetics. In this respect, this is a timely and important study sure to be stimulating to the community. I therefore offer my recommendation for publication.

We thank the reviewer for a detailed and accurate summary of our work and its significance. Thanks also for making suggestions for improvement. We have implemented both of your suggestions, as described below.

Minor suggested revisions: 1. In the SI, just before (S2), could the authors justify their use of the “long time approximation” here.

We had not been clear enough. Since torque production is driven by proton passage through the stator units (which is discontinuous), the actual torque produced by the stator units varies with time. However, over a time scale that is much longer than the time scale of proton translocation, torque may be assumed constant to arrive at the analytical result given in equation S2.

To clarify, we have revised the text to: “Over a time scale much longer than the time sale of proton translocation through the stator units, Γ may be considered constant. We can then solve...”

2. In the SI, just before (S10) the authors set $k_{\text{off},t} = 0$. This confused me at first, since in the main text, it is not zero, merely very small. A suggested edit, would be to clarify that by setting $k_{\text{off},t} = 0$, it is the first step in an approximation where it is small, and not identically zero to align better with the main text.

Thank you for this suggestion. We have revised the text to: “Since a stator unit has a much smaller off rate in the **T** state as compared to the **L** state, i.e., $k_{\text{off},t} \ll k_{\text{off},l}$, by setting $k_{\text{off},t} = 0$ we can approximate the overall off rate k_- as:”

References

1. Guo, S., Xu, H., Chang, Y., Motaleb, M. A. & Liu, J. Flil ring enhances the function of periplasmic flagella. *Proc. Natl. Acad. Sci.* **119**, e2117245119 (2022).
2. Tachiyama, S. *et al.* The flagellar motor protein flil forms a scaffold of circumferentially positioned rings required for stator activation. *Proc. Natl. Acad. Sci.* **119**, e2118401119 (2022).

REVIEWERS' COMMENTS

Reviewer #2 (Remarks to the Author):

The authors performed successfully the additional statistical tests and clarifications required.